# Circulating miRNA Increases the Diagnostic Accuracy of Chromogranin A in Metastatic Pancreatic Neuroendocrine Tumors

**DOI:** 10.3390/cancers12092488

**Published:** 2020-09-02

**Authors:** Annamária Kövesdi, Petra Anna Kurucz, Gábor Nyírő, Ottó Darvasi, Attila Patócs, Henriett Butz

**Affiliations:** 12nd Department of Internal Medicine, Semmelweis University, 1088 Budapest, Hungary; ancsa.kovesdi@gmail.com; 2Department of Laboratory Medicine, Semmelweis University, 1089 Budapest, Hungary; kurucz.petra1997@gmail.com (P.A.K.); butz.henriett@med.semmelweis-univ.hu (H.B.); 3Molecular Medicine Research Group, Hungarian Academy of Sciences and Semmelweis University, 1088 Budapest, Hungary; nyirogabor1@gmail.com; 4Hereditary Tumours Research Group, Hungarian Academy of Sciences and Semmelweis University, 1089 Budapest, Hungary; otto.darvasi@gmail.com; 5Department of Molecular Genetics, National Institute of Oncology, 1122 Budapest, Hungary

**Keywords:** circulating miRNA, biomarker, neuroendocrine tumor, pheochromocytoma, pancreas, pNET

## Abstract

**Simple Summary:**

Despite its varying sensitivity and decreased specificity, chromogranin A (CgA) is the most widely used biomarker for neuroendocrine tumors. The most common factor affecting its diagnostic accuracy is the use of proton pump inhibitors (PPIs). Our aim was to investigate circulating miRNA expression profiles in pancreatic neuroendocrine tumors (pNET) and pheochromocytomas/paragangliomas (PPGL) to find miRNAs which could be used as biomarkers along with CgA in these patients. MiRNA expression profiles were determined with next generation sequencing and validated by quantitative real time PCR in 74 samples obtained from patients and healthy volunteers treated with PPI. We observed a global downregulation of miRNAs in NET compared to controls. A set of miRNAs in combination with CgA resulted in the best discrimination of pNET irrespective of PPI treatment and a combination of miRNAs increased the diagnostic utility of CgA even in pNET patients with low CgA.

**Abstract:**

Chromogranin A (CgA) is the most widely accepted biomarker for neuroendocrine tumors (NET) but its diagnostic accuracy is dependent on tumor type and the use of proton-pump inhibitors (PPI). We investigated the diagnostic value of circulating miRNAs along with CgA in pancreatic neuroendocrine tumors (pNET). 74 serum samples from patients with pNET (*n* = 25, nonfunctioning), pheochromocytoma/paraganglioma (PPGL, *n* = 20), healthy individuals with normal CgA (*n* = 29) including 10 samples from 5 healthy individuals with and without current PPI treatment were collected. MiRNA expression profiles were determined using next-generation sequencing, followed by validation with individual TaqMan assays. A global downregulation of miRNAs was observed in patients with NET compared to controls. MiRNA expression of 33 miRNAs was able to discriminate tumor samples from controls. No miRNA alone could be considered as an applicable biomarker for pNET or PPGL. However, using a logistic model, the combination of a set of miRNAs increased the discriminatory role of CgA irrespective of PPI treatment. In pNET patients with normal CgA level our regression model yielded high (89.4%) diagnostic accuracy (AUC: 0.904, sensitivity: 66.6%, specificity: 96.5%). A set of miRNAs increased the diagnostic utility of CgA in pNET even in patients with low CgA.

## 1. Introduction

Chromogranins are polypeptide prohormones being the major constituents of dense-core secretory granules in neuroendocrine cells and they are co-secreted with peptide hormones and amines [1].

Neuroendocrine tumors (NETs) originate from neuroendocrine cells distributed widely throughout the body [2]. They are rare neoplasms but their incidence and prevalence are steadily rising [3]. Pancreatic NET (pNET) is the most common neuroendocrine tumor that can either be functioning or nonfunctioning according to hormone production [2]. Nonfunctioning (NF) pNETs comprising the largest group of pNETs being asymptomatic result in delayed diagnosis and often present as metastatic at diagnosis that significantly affect prognosis and survival [2,4].

An elevated level of circulating CgA has been associated with almost all tumor types of the neuroendocrine system [1,5], but its sensitivity varies between 47–100% depending on tumor type (100%, in gastrinomas, ~89% in pheochromocytomas, and ~69% in nonfunctioning pNETs) [1,6,7,8]. In addition, many oncological and non-oncological conditions, including renal failure, non-endocrine tumors (e.g., prostate, breast, thymus, uterus, colon), chronic gastritis, and current proton pump inhibitor therapy (PPIs) decrease the specificity CgA [6,9], but CgA is the only routinely measured circulating marker in pNET. Previous studies, including one from our research group, showed that both short- and long-term applications of PPIs and other acid-suppressive medications result in a significant increase of CgA in blood, thus decreasing its clinical utility in the management of NETs [10,11].

Neuron-specific enolase (NSE) is a cytosolic enzyme of neurons and neuroendocrine cells, that is predominantly expressed in small-cell lung cancer and poorly differentiated NETs [12,13]. Its diagnostic value, particularly its specificity for neuroendocrine tumors is rather poor [8,14,15]. 5-Hydroxyindoleacitic acid (5-HIAA), as the main metabolite of serotonin, is a useful biomarker of NETs in patients with carcinoid syndrome [12]. Other biomarkers, such as insulin, glucagon, or gastrin are specific for subtypes of neuroendocrine cells [12].

Pheochromocytomas/paragangliomas (PPGLs) are rare neuroendocrine tumors of chromaffin cell origin. These are found in the adrenal medulla (pheochromocytomas, PCC) and the paravertebral ganglia of the nervous system (paragangliomas, PGL) [16]. They can produce catecholamines, but PGLs located above the diaphragm are often non-secretory [16]. In PPGL, as CgA is co-secreted with catecholamines, besides the determination of catecholamines and their metabolites, elevated CgA may indicate tumor mass and malignancy; therefore, it may be used to monitor clinical response and tumor relapse especially when the measurement of plasma metanephrines is not accessible [16].

Similarly to gene expression, miRNA expression is also cell type-specific and due to their presence in circulation and their stability, they are considered as promising biomarker candidates in several tumor types [17,18]. Their previously described expression alterations in pNET and PPGL raise their potential usefulness as circulating biomarkers [19,20]. A multianalyte biomarker (NETest) encompassing 51 different NET-related transcripts showed a significantly more sensitive and efficient method in NET diagnosis compared to single biomarkers [21], however, its availability and costs may limit its usefulness [12].

We aimed to investigate the serum miRNA expression profile in patients with pNET to identify potentially useful miRNA biomarkers. PPGL and healthy individuals with and without PPI treatment were also studied to challenge the diagnostic utility of identified miRNAs along with CgA.

## 2. Results

### 2.1. Evaluation of Cell Contamination Based on miRNA Signature

As intracellular miRNA concentration is much higher compared to those measured in a cell-free environment, cellular contamination is an important factor to avoid. Therefore, first, we assessed whether cellular contamination in our serum samples was present. In the collected samples visually, no sign of hemolysis could be seen. The ratio of red blood cell-enriched miR-451a to the reference miR-23a-3p was revealed as the most sensitive method for the detection of hemolysis compared to hemoglobin measurement techniques (Coulter^®^ AcT diff Analyzer and spectrophotometric method) [22]. Assessing miR-451a to miR-23a-3p ratio no red blood cell (RBC) contamination was detected in any of our samples.MiRNAs which are considered indicating red-blood-cell or platelet cellular contamination [23,24,25] also showed low variance (0.4 for miR-142-3p and 0.3 for miR451a/miR-23a-3p, respectively). Only samples without cellular contamination were included.

### 2.2. Analysis of Circulating miRNAs by Next-Generation Sequencing

Our exploratory study included 24 samples: 8 pNET (4 patients with low- and 4 with high-CgA), 8 PPGL patients and 8 control samples from 6 healthy individuals (2 of them were also treated with PPI, samples before and after PPI treatment were both included). Altogether, 1525 miRNA were detected in NGS profiling of whole serum samples. While nine miRNAs were the most abundant with an >500 UMI reads/sample representing 0.05% of all detected miRNAs, the great majority of miRNAs (1461 of 1525, 95%) were in the low-expression range (below 50 UMI reads/sample) (Figure 1A,B). Total (global) miRNA expression in patients with pNET or PPGL was significantly decreased compared to normal controls (Figure 1C).

Low abundance (<50 UMI read number) miRNAs were excluded from further analysis because these miRNAs could not be validated in our experimental set up [26]. 33 miRNAs differentiated pNET/PPGL patients from normal controls irrespective of PPI treatment (Table 1, Figure 1D). In pNET 19 and 7 under- and over-expressed miRNA, while in PPGL 25 and 6 under- and over-expressed miRNAs compared to healthy controls were found (Table 1). However, PNET and PPGL patients could not be differentiated by serum miRNA expression (Figure 1D). Additionally, we did not find significant differences in healthy controls before and after PPI treatment.

Expression of 19 miRNAs showed a significant correlation with serum CgA levels among which hsa-let-7g-5p showed the strongest association (R = −0.73; *p* = 0.03077).

### 2.3. miRNA Validation

Of the 33 significantly differently expressed miRNAs between controls and tumor sera samples we selected 6 underexpressed miRNAs (let-7b-5p, let-7i-5p; miR-143-3p; miR-30d-5p; miR-451a; miR-486-5p) in pNET/PPGL samples compared to controls) based on miRNA abundance for further investigation. Additionally, we also validated the most overexpressed miR-203a-3p despite its relatively lower abundance (avg. UMI read number in pNET/PPGL samples: 68 vs. avg. UMI read number in controls: 19). The selected miRNAs were measured on an extended sample cohort consisting of 25 pNET and 20 PPGL patients (together with those included in the exploratory cohort), and 29 control samples (5 with PPI treatment, 24 without PPI treatment; including samples from our exploratory cohort).

All the 6 underexpressed miRNAs were confirmed by RT-qPCR as well (Figure 2A). After dissecting tumor groups, the underexpression of miR-30d-5p, miR-451a, and miR-486-5p was significant in pNET patients with high CgA levels and let-7b-5p in PPGL cases compared to healthy controls without PPI treatment (Figure 2B).

The overexpression of miR-203a-3p was not confirmed by RT-qPCR in any of the samples but one pNET patient with extremely high CgA (2490.8 ng/mL).

### 2.4. Association of miRNA Expression with Clinicopathological Parameters

We could not find a difference in miRNA expression profiles of pNET/PPGL samples compared to healthy controls treated with PPI. However, let-7b-5p, let-7i-5p; miR-143-3p; miR-30d-5p; miR-451a and miR-486-5p showed lower expression in samples with high (>100 ng/mL) CgA levels compared to samples with normal CgA level (≤100 ng/mL) (Figure 3A). The expression of all 6 miRNAs showed a negative correlation with serum CgA level when all samples were included in the analysis. While investigating correlation in different patient groups the negative correlation between miRNAs and CgA remained significant only in the healthy control group but not in pNET or PPGL group (Table 2).

Investigating miRNAs in relation to the tumor grade, in pNET patients a tendency of inverse correlation between miRNAs: let-7i-5p, miR-30d-5p, and miR-451a and grade was observed (Figure 3B).

In samples obtained from PPGL patients only miR-486-5p showed an inverse correlation with CgA (Table 2). As different miRNA signatures have been described in malignant vs. benign [27] and sporadic vs. hereditary PPGL [28] we aimed to investigate serum miRNAs in correlation to these clinicopathological parameters. We found that in serum samples obtained from PPGL patients with germline mutations (*SDHB*, *RET*, *VHL,* or *NF1*) miR-486-5p and miR-30d-5p were downregulated compared to sporadic patients (Figure 3C). In patients with *SDHB* germline mutation beside the underexpression of miR-30d-5p and miR-486-5p, let-7b-5p and let-7i-5p also showed downregulation compared to serum samples of patients with *SDHB* wild type (Figure 3D).

### 2.5. Diagnostic Value of the Circulating miRNAs in PPGL/pNET

In the discrimination of healthy controls from pNET and PPGL patients, we performed receiver-operating characteristic (ROC) analysis for CgA and individual miRNAs. Binary logistic regression models were generated for combinations of miRNAs and combinations of CgA with miRNAs. Altogether 6 miRNAs (let-7b-5p, let-7i-5p, miR-143-3p, miR-30d-5p, miR-451a, and miR-486-5p) were assessed individually and in combinations (Table 3). In the pNET group compared to CgA the use of individual let-7b-5p, miR-30d-5p, miR-451a, and miR-486-5p in ROC analysis resulted in better discrimination (Table 3). Additionally, the regression model of the combination of CgA with a panel of 4 (let-7b-5p, let-7i-5p, hsa-miR-143-3p, miR-30d-5p) miRNAs resulted in the best discrimination with 72.2% accuracy (AUC: 0.751; 75.8% sensitivity; 68% specificity) (Table 3).

Clinically, the most challenging pNET patients are those with normal serum CgA levels (<100 ng/mL). Therefore, miRNA expression profiles are of particular interest in these patients. The use of the individual expression of let-7b-5p or miR-143-3p yielded better discrimination from healthy controls compared to CgA alone (Table 3). However, the combination of CgA value with an expression of 4 miRNAs in a logistic regression model showed the best discrimination between pNET-low CgA group compared to healthy controls including normal or elevated CgA with the accuracy of 89.4% (AUC: 0.904, sensitivity: 66.6%, specificity: 96.5%) (Table 3).

In our sample set all PPGL patients had elevated CgA level, and in line with this CgA, had the highest discriminatory value with 81.6% accuracy (AUC: 0.889; 75% sensitivity, 86% specificity; *p* = 0.0007). Using binary logistic regression model the combination of CgA with 3 miRNAs (let-7b-5p, miR-143-3p and miR-486-5p) yielded higher specificity compared to CgA alone with 83.6% accuracy in discrimination of normal samples from PPGL samples, however, with lower sensitivity (AUC: 0.862; 85% sensitivity; 82.7% specificity) (Table 3).

### 2.6. Analysis of Deregulated pNET/PPGL Tissue miRNAs in Serum Samples

To investigate the potential origin of miRNAs in serum, we collected the data of 12 studies reporting tissue miRNA expression profiles in pNET [29,30,31,32,33,34,35] and PPGL [27,28,36,37,38]. By cross-referencing differentially expressed miRNAs from those studies with our current findings 9 miRNAs in pNET and 3 miRNAs in PPGL were found commonly dysregulated in both tissue and serum. (Table 4). Among these, only miR-203a-3p was overexpressed both in pNET tissue and serum. All other miRNAs were upregulated in tumor tissues and downregulated in serum samples. In PPGL, while miR-101-3p showed higher expression in malignant vs. benign PPGLs, in serum, it was downregulated compared to controls. However, miR-16-5p and miR-451a as putative tumor suppressor miRNAs were downregulated in both tissue and serum (Table 4).

## 3. Discussion

Tumors of the neuroendocrine system are rare entities but represent serious diagnostic and therapeutical challenges. Especially, when they are hormonally inactive, the diagnosis is usually delayed due to a lack of specific circulating biomarkers. The delayed diagnosis influences prognosis and survival [2]. Therefore, novel circulating biomarkers are needed. To our best knowledge, no study on the diagnostic accuracy of circulating miRNAs in neuroendocrine tumors in relation to serum CgA has been published.

Our data revealed a low global miRNA expression (abundance) in sera of patients with pNET/PPGL compared to normal controls. The same phenomenon was also observed in blood samples obtained from patients with pituitary adenoma [26]. Interestingly, in tumor tissues, a general downregulation of miRNAs compared to normal tissues was also detected suggesting their general tumor-suppressive role instead of being oncogenes [39]. Based on the individual miRNA measurements, Mitchell et al., 2008 [17] reported that the presence of tumors does not lead to a generalized increase in circulating miRNAs. However, specific individual tumor-derived miRNAs can show increased expression in circulation. With the technical advancement of NGS, total miRNA abundance can be determined easier and more accurately than earlier used platforms (i.e., microarray) allowed. Still, both the causes and the role of the decreased global miRNA expression in circulation related to endocrine tumors are waiting to be further explored.

The origin of circulating miRNAs is still not clearly defined [40]. The high-abundance cell-free circulating miRNAs derive primarily from blood and endothelial cells, and tumor-derived miRNAs are possibly among the moderate or low abundance range which highlights the importance of the accurate detection of moderate-low abundance miRNAs [17,24,40]. The overexpressed miRNAs can originate directly from tumor tissues [17]. By cross-referencing these with the tissue miRNA expression we identified 3 miRNAs commonly dysregulated in both tissue and circulation. MiR-203a-3p was overexpressed both in pNET tissue and serum. Similarly, to our findings, in breast cancer, the same miR-203a-3p was overexpressed both on a tissue level and in exosomes secreted by tumor cells [41,42]. Functionally, it exhibits oncogenic effects in colon and breast cancer, and through MYC and it influences taxane sensitivity [41,43,44]. These data raise the possibility that miR-203a-3p may potentially be originated from tumor cells and can be considered as an oncomiR. However, its low abundance in the sera of patients with NET limits its wide clinical applicability as a tumor marker. Among downregulated miRNAs, miR-16-5p and miR-451a were both downregulated in PPGL tissues and sera indicating their roles as potential tumor suppressors that are well-established in other cancers [45,46].

Based on the expression pattern of 33 miRNAs we were able to separate pNET/PPGL patients from normal controls even on PPI treatment. Based on serum miRNA expression we could not distinguish samples of pNET from those of PPGL patients. This may reflect the common enterochromaffin cell type origin of these tumors [47].

Based on NGS data, several miRNAs showed (mostly inverse) correlation with serum CgA level. However, after validation on an extended sample set, this correlation remained significant only in healthy individuals but not in pNET or PGGL patients (except for miR-486-5p in PPGL patients). This may be due to the loss of the great majority of miRNAs showing correlation with CgA during tumorigenesis (as the global expression of miRNAs decreased in patients compared to healthy individuals) or due to technical issues. NGS allows us to detect any miRNA with low or very low copies but RT-qPCR has a detection limit, in our experimental settings miRNAs with reads <50 UMI could not be validated with RT-qPCR.

Increased CgA level upon PPI treatment decreases the clinical utility of CgA in neuroendocrine tumors. Performing ROC analysis, we found that in pNET samples CgA alone was less effective in discrimination of pNET sera from healthy controls. The addition of miRNAs to CgA increased the diagnostic value of CgA. Importantly, in patients with pNET with normal CgA level where CgA alone was not informative, the combination of CgA with a miRNA panel could reach high diagnostic value (AUC: 0.904).

In pNET patients, the grade, determined by the Ki-67 proliferation index, has an important impact on prognosis [48]. Although the low sample number represents a limitation of our study, the expression of miR-451a, let-7i-5p, and miR-30d-5p showed a tendency of an inverse correlation with a grade in pNET patients. Circulating miR-451a was reported underexpressed in several tumor types and it was described as an immune-regulatory miRNA [45,49,50,51] that may explain its dysregulation according to tumor grade in pNET. Let-7i-5p is a well-known tumor suppressor miRNA which expression is negatively correlated with prognosis [52]. In several studies, it was underexpressed in sera of cancer patients vs. healthy controls [52] but the explanation behind its decreasing expression and its role in pNET patients with higher-grade remains to be clarified. MiR-30d-5p was also reported downregulated in tumor (hepatocellular carcinoma) cells and secreted exosomes and its high expression in hepatocellular carcinoma exhibited improved overall survival [53]. These data suggest that miR-30d-5p, measured both in tissue and in liquid biopsy samples, may indeed have a role in relation to tumor grade, progression, and survival.

Previously, in pNET, only one study investigated circulating miRNA levels compared to healthy controls [30]. The authors found a very similar miRNA expression profile compared to healthy volunteers [30]. Only miR-193b was described to be more abundant in the serum of patients and it was also more abundant in pNET tissue compared to islet cells. However, by analyzing both tissue and serum samples of 6 patients, there was a much lower correlation between expression levels in serum and tissue compared to the mean correlation coefficient within the same tissue [30]. In another study, 5 miRNAs were identified differentially expressed in the sera of pancreatic ductal adenocarcinoma patients compared to pNET but no comparison between pNET and healthy controls was performed [54]. None of these 5 miRNAs showed overlap with our results comparing pNET vs. controls. In a very recent publication, 6 upregulated and 11 downregulated miRNAs were identified in plasma exosome fraction of pNET patients (no functionality reported) compared to chronic pancreatitis [55]. Again, none of the identified miRNAs was identical to those in our study although they had different starting material, different measurement methods, and most importantly, they compared their data to chronic pancreatitis.

The decrease of miR-486-5p and miR-30d-5p was detected in patients with genetically determined PGGLs (mutation in either *SDHB*, *RET*, *VHL*, or *NF1* genes) compared to sporadic cases. Let-7b-5p, let-7i-5p, miR-30d-5p, and miR-486-5p were downregulated in *SDHB* mutant cases compared to *SDHB* wild type cases indicating that the etiology of PPGL may influence the circulating miRNA pool. This observation needs further studies because no data about serum and tissue miRNA differences between sporadic and genetic PPGL cases were found. Four studies investigated circulating miRNAs in PPGLs [27,36,56,57]. No significant difference was demonstrated between benign and malignant cases but elevated levels of miR-96-5p; miR-182-5p and miR-21-3p in PPGL patients compared to healthy controls were reported [27,36,56]. We could not confirm these results probably due to differences in experimental design (mirVana™ PARIS™ Kit vs. miRNeasy Serum Kit for isolation; TaqMan assays vs. QIAseq™ miRNA Library Kit and droplet digital PCR vs. next-generation sequencing).

As a summary, we assessed for the first time the diagnostic accuracy of circulating miRNAs in relation to CgA in patients with pNET. Using the miRNA profile, we were able to discriminate pNET/PPGL samples from healthy controls, and a global downregulation of circulating miRNA pool was observed in patients with neuroendocrine tumors compared to healthy individuals. Similarly, to CgA, miRNA signature alone was not applicable as a biomarker for PPGL/pNET if PPI treatment is administered. However, the combination of a set of miRNAs together with CgA, even in the most challenging group (pNET patients with low CgA level or receiving PPI treatment) represents a promising diagnostic tool.

## 4. Materials and Methods

### 4.1. Patients and Controls

Altogether, 74 consecutive serum samples sent for CgA measurement to the Department of Laboratory Medicine, Semmelweis University were collected (Table 5) between 1 November 2017 and 24 July 2019. Samples were obtained from patients with pancreas nonfunctioning neuroendocrine tumor (pNET, *n* = 25), with pheochromocytoma-paraganglioma (PPGL, *n* = 20) and from healthy individuals (*n* = 29 samples from 24 unrelated cases). PPGL and healthy individuals treated with PPI were included as controls because in clinical practice elevated CgA levels most often can be found in these cases.

All patients with pNET (12 females, 13 males; avg age: 63.4 years) had metastatic disease. Low-CgA (<100 ng/mL) pNET group consisted of 9, high-CgA (>100 ng/mL) pNET group consisted of 16 patients. Grade of pNET was determined according to WHO 2017 classification of tumors of endocrine organs based on the Ki-67 index (Table 5) [58].

PPGL group consisted of 12 females and 8 males with an average age of 41.8 years. Of them, 8 were benign, 11 were malignant (1 bilateral, 1 locally receive, and 9 metastatic) and there was no information available in one case. 14 of the 20 PPGL cases were sporadic, while 6 cases were associated with germline gene mutations in one of the PPGL susceptibility genes (Table 5). All PPGL patients had elevated CgA levels (avg. CgA: 1751 ng/mL).

Altogether 24 healthy controls were included in the study. Among 29 control samples, 10 sera from 5 healthy volunteers were collected with and without proton pump inhibitor (PPI) treatment. Samples from these individuals were included if high-dose PPI at least for 2 weeks were administered [11]. In our discovery analysis, we used 8 control samples from 6 healthy individuals, 2 samples were taken from two cases after PPI treatment. None of the controls had the endocrine disease, tumor, or renal insufficiency. Eight samples obtained from patients with pNET and 8 samples obtained from PPGL patients were evaluated. In our validation set, we used all samples used in the discovery analysis along with 17 additional samples obtained from pNET, 12 samples obtained from PPGL patients, and 21 samples obtained from healthy controls (Table 5). Clinical information was retrieved from the Semmelweis University medical information system. The study was approved by the Hungarian National Public Health Center (NPHC: 41189-7/2018/EÜIG, 13 December 2018) and the Scientific and Research Committee of the Medical Research Council of Ministry of Health, Hungary (ETT-TUKEB 4457/2012/EKU, 2 February 2012).

### 4.2. Sample Handling and RNA Extraction

Upon arriving samples were immediately processed for sera separation by centrifugation with 3000× *g* for 10 min at 4 °C. Then, samples were aliquoted and refrigerated at −20 °C until further processing. Samples for RNA isolation were further processed by centrifugation with 16,000× *g* for 15 min at 4 °C to deplete platelets.

CgA measurements were done as part of the routine diagnostics using an IVD qualified competitive radioimmunassay method (Chromogranin A kit; REF: CGA-RIACT; Cisbio Bioassays, Codolet, France) following the manufacturer instructions on RIA-mat-280 gamma counter (Byk-Sangtec Diagnostica, Dietzenbach, Germany). 

For total RNA extraction containing small RNA fraction miRNeasy Serum/Plasma Kit (Qiagen, Cat #217184, Hilden, Germany) was applied. Custom cel-miR-39 spike-in control was added to samples during RNA extraction following the manufacturer’s instruction (Qiagen, Cat # 219610) [26]. Cellular contamination of serum samples was studied by evaluation of expression of miRNAs related to red blood cells and platelets.

### 4.3. miRNA Expression Profiling by Next-Generation Sequencing

MiRNA expression profiling and NGS data analysis were done as we earlier reported [26,59]. For miRNA sequencing, libraries were prepared using 5 ul RNA and the QIAseq™ miRNA Library Kit (Qiagen, Cat #331505) following the manufacturer’s instructions. Next-generation sequencing was run on Illumina MiSeq instrument using MiSeq Reagent Kit v3 150-cycle (MS-102-3001, Illumina, San Diego, CA, USA. For count normalization TMM was applied as it is one of the best performing algorithm for comprehensive miRNA profiling studies [60].

### 4.4. RT-qPCR Validation of NGS Results 

For validation TaqMan Advanced miRNA cDNA Synthesis Kit (Cat#: A28007, Thermo Fisher Scientific, Waltham, MA, USA) was applied. The expression of miRNAs was determined using individual Advanced TaqMan MicroRNA Assays (Assay IDs: cel-miR-39-3p: 478293_mir, hsa-let-7b-5p: 478575_mir; hsa-let-7i-5p: 478375_mir; hsa-miR-143-3p: 477912_mir; hsa-miR-203a-3p: 478316_mir; hsa-miR-30d-5p: 478606_mir; hsa-miR-451a: 478107_mir; hsa-miR-486-5p: 478128_mir). Expression levels were calculated by the ddCt method, and fold changes were obtained using the formula 2^-ddCt^. All measurements were done in triplicates.

### 4.5. Literature Mining for Tissue-Serum miRNA Expression Cross-Referencing

To investigate the potential origin of serum miRNAs and to compare serum and tissue miRNA expression profiles, we collected data of miRNA expression profiling studies performed on pNET (7 studies) [29,30,31,32,33,34,35] and PPGL (5 studies) [27,28,36,37,38] by PubMed search. We cross-referenced our findings with data retrieved from these studies.

### 4.6. Statistical Analysis

Shapiro-Wilks test was applied to reveal data distribution. A comparison of differential expression was done by unpaired T-test or Mann–Whitney U test and one-way ANOVA followed by Tukey post-hoc test depending on data distribution and grouping. P-value adjustments were applied by Benjamini–Hochberg. For hierarchical cluster analysis, complete linkage clustering with Kendall’s Tau distance measuring method was used. The receiver–operator characteristic (ROC) analysis of CgA and miRNAs’ were used for their discriminatory role between different groups. Binary logistic regression analyses were applied to evaluate the relationships between miRNAs and the incidence of PPGL/pNET and to find the best logistic model. Diagnostic accuracy was calculated as: Accuracy = (TN + TP)/(TN + TP + FN + FP) = (Number of correct assessments)/(Number of all assessments). (TN: true negative; TP: true positive; FN: false negative; FP: false positive).

*p*-Value < 0.05 was considered significant. Statistical analyses were done using the R statistical programming language.

## 5. Conclusions

As a summary, we assessed for the first time the diagnostic accuracy of circulating miRNAs in relation to CgA in patients with pNET. Using the miRNA profile, we were able to discriminate pNET/PPGL samples from healthy controls, and a global downregulation of circulating miRNA pool was observed in patients with neuroendocrine tumors compared to healthy individuals.

In a comparison of pNET or PPGL groups with controls, levels of CgA, and miRNAs showed significant differences. If samples from controls treated with PPI were included either CgA or miRNA signature alone was not applicable as biomarker for PPGL/pNET. In addition, neither CgA nor any miRNA alone was able to discriminate low CgA pNET patients from controls treated with PPI. However, the combination of CgA concentration and expression level of miRNAs in a regression model yielded higher AUC compared to those obtained using individual markers, demonstrating that our model was effective in distinguishing different groups in every comparison. Our results are in line with previous findings showing that in other cancer types, independently of individually being significantly different or not, combining more factors in a regression model increases the discriminatory value of biomarkers [61,62]. Therefore, the combination of a set of miRNAs together with CgA, even in the most challenging group (pNET patients with low CgA level or receiving PPI treatment) represents a promising diagnostic tool.

## Figures and Tables

**Figure 1 cancers-12-02488-f001:**
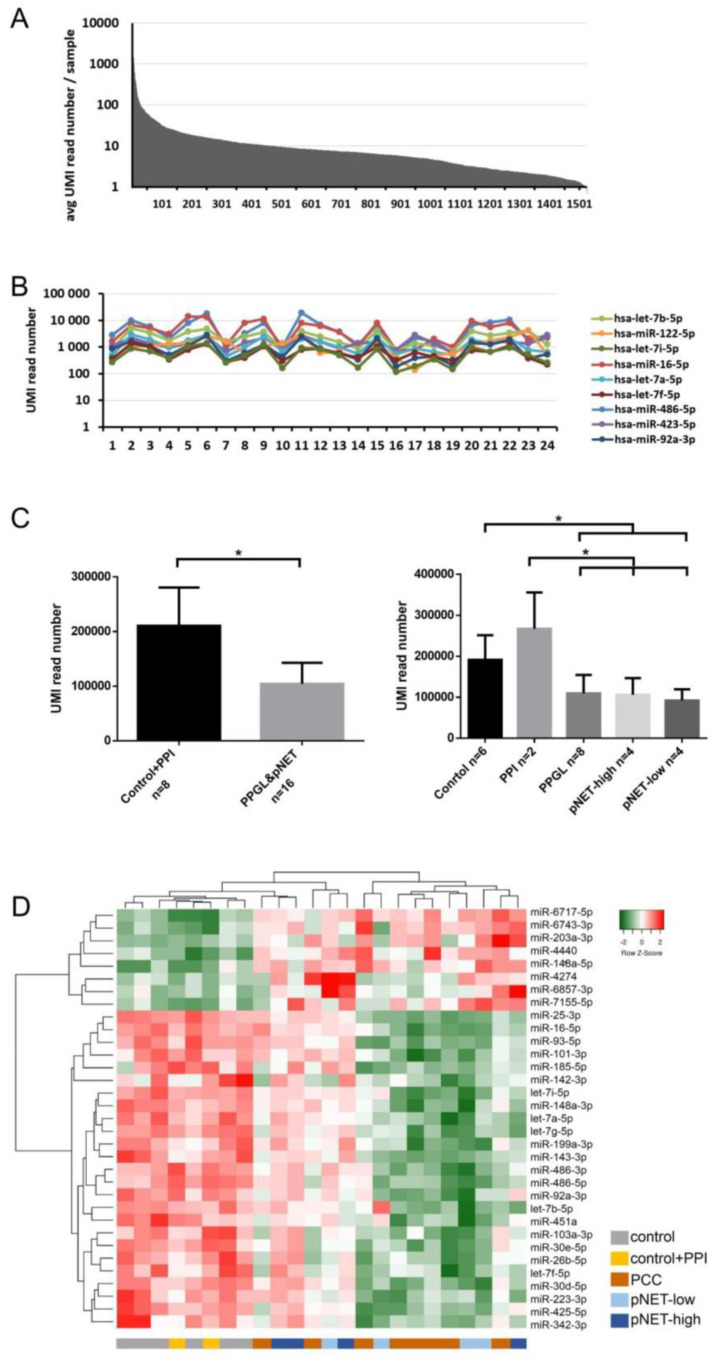
MiRNA expression in the exploratory study. (**A**) MiRNA abundance in serum. (**B**) Expression of the 9 most abundant miRNAs. (**C**) Global miRNA expression level among normal controls and pNET/PCC samples. Error bars indicate mean ± SE. * represents *p*-value < 0.05. (**D**) 33 differentially expressed miRNAs among different groups can discriminate against control and pNET/PCC samples. Green indicates lower, red indicates higher expression.

**Figure 2 cancers-12-02488-f002:**
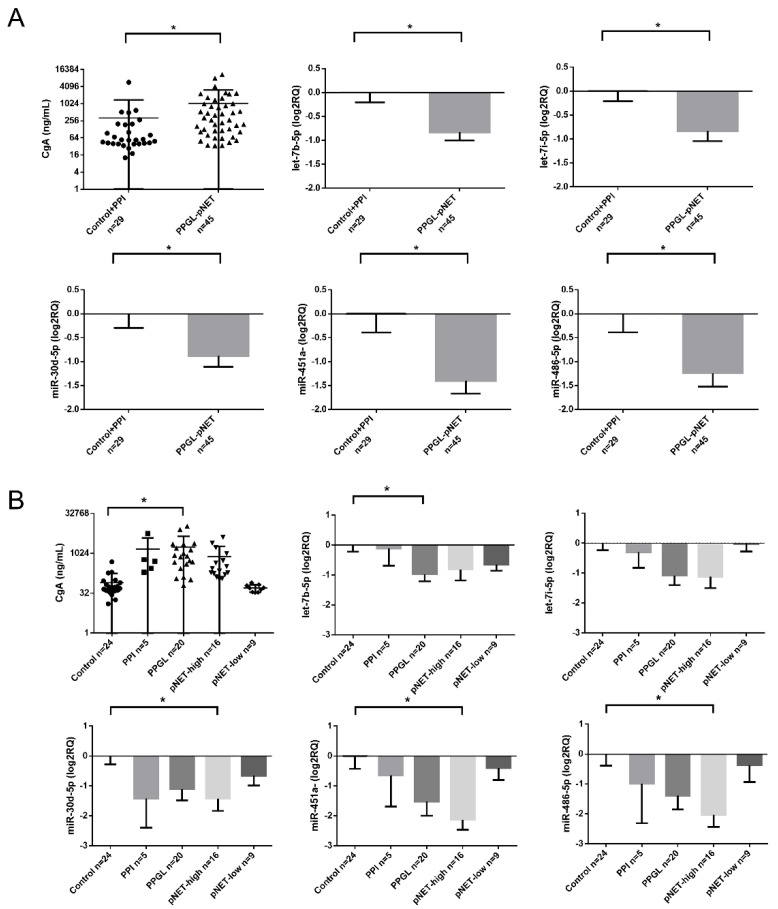
(**A**,**B**) CgA levels and miRNA expression among controls and pNET/PPGL samples. Error bars indicate mean ± SE. * represents *p*-value < 0.05.

**Figure 3 cancers-12-02488-f003:**
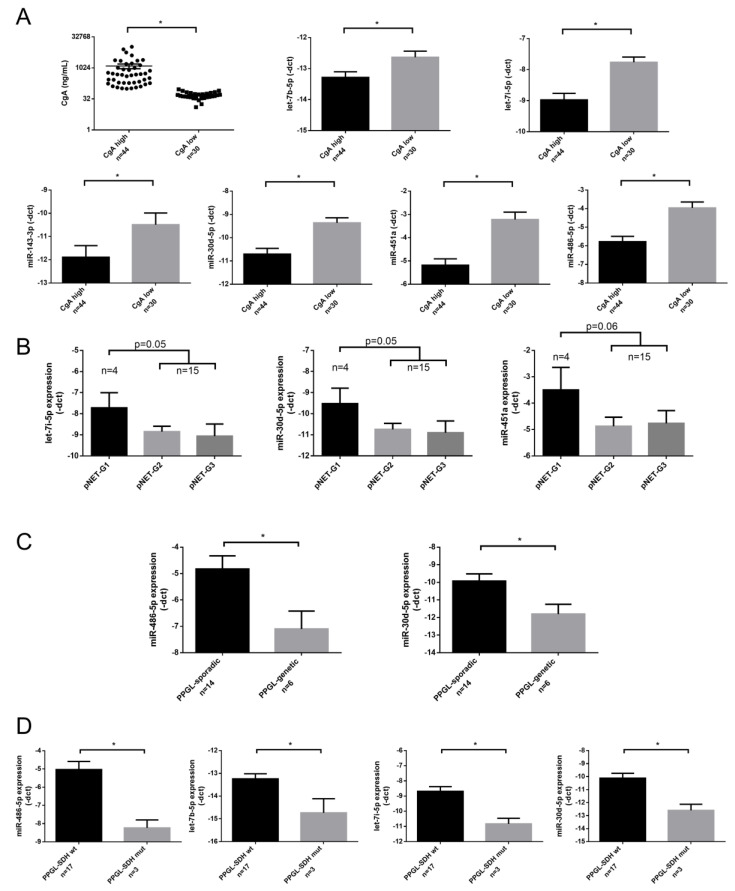
Individual qPCR validation of miRNAs in CgA high vs. CgA low samples (**A**), among pNET samples with different grade (**B**), in sporadic vs. genetically predisposed PPGL (**C**), and in *SDHB* mutation associated vs. non-associated PPGL (**D**). Error bars indicate mean ± SE. * represents *p*-value *<* 0.05.

**Table 1 cancers-12-02488-t001:** 33 miRNAs differentially expressed among controls with or without PPI treatment vs. PPGL vs. pNET groups. (“control&PPI” group includes all healthy controls with and without PPI treatment: “tumor-free” healthy individuals irrespective of the usage of PPI).

miRNA	ANOVA *p*-Value	pNET vs. Control&PPI		PGGL vs. Control&PPI		pNET vs. PPGL	
		log2FC	Post-hoc *p*-Value	log2FC	Post-hoc *p*-Value	log2FC	Post-hoc *p*-Value
hsa-miR-223-3p	0.0015	−1.62	0.0304	−2.27	0.0235	0.65	1
hsa-miR-486-3p	0.0002	−1.56	0.0177	−2.51	0.0066	0.95	1
hsa-miR-451a	0.0003	−1.42	0.0257	−2.35	0.0058	0.93	1
hsa-miR-16-5p	0.0009	−1.48	0.0320	−2.00	0.0139	0.52	1
hsa-miR-25-3p	0.0001	−1.48	0.0153	−2.10	0.0054	0.62	1
hsa-miR-143-3p	0.0001	−1.35	0.0240	−2.34	0.0031	1.00	1
hsa-miR-101-3p	0.0040	−1.31	0.0823	−1.90	0.0273	0.60	1
hsa-miR-486-5p	0.0005	−1.27	0.0413	−2.50	0.0066	1.23	1
hsa-miR-425-5p	0.0000	−1.23	0.0178	−2.23	0.0024	0.99	1
hsa-miR-148a-3p	0.0010	−1.25	0.0322	−1.82	0.0146	0.57	1
hsa-miR-93-5p	0.0003	−1.29	0.0240	−1.97	0.0066	0.68	1
hsa-let-7g-5p	0.0000	−1.14	0.0178	−2.20	0.0024	1.06	1
hsa-let-7i-5p	0.0003	−1.06	0.0316	−1.89	0.0055	0.83	1
hsa-miR-92a-3p	0.0004	−1.05	0.0440	−2.05	0.0060	1.01	1
hsa-let-7a-5p	0.0003	−1.00	0.0235	−1.66	0.0064	0.66	1
hsa-miR-185-5p	0.0008	−1.02	0.0859	−2.23	0.0079	1.22	1
hsa-miR-342-3p	0.0029	−0.97	0.0624	−1.63	0.0249	0.66	1
hsa-miR-30e-5p	0.0001	−0.93	0.0295	−1.77	0.0031	0.84	1
hsa-miR-142-3p	0.0055	−0.85	0.1610	−1.71	0.0287	0.86	1
hsa-miR-30d-5p	0.0015	−0.88	0.0432	−1.57	0.0165	0.69	1
hsa-let-7b-5p	0.0029	−0.84	0.0539	−1.51	0.0261	0.67	1
hsa-let-7f-5p	0.0007	−0.75	0.0475	−1.41	0.0081	0.66	1
hsa-miR-199a-3p	0.0069	−0.77	0.1009	−1.47	0.0375	0.70	1
hsa-miR-103a-3p	0.0003	−0.74	0.0371	−1.38	0.0055	0.63	1
hsa-miR-26b-5p	0.0003	−0.65	0.0424	−1.31	0.0054	0.66	1
hsa-miR-6743-3p	0.0002	0.61	0.0179	0.45	0.0054	0.15	1
hsa-miR-4440	0.0000	0.65	0.0107	0.54	0.0031	0.11	1
hsa-miR-6717-5p	0.0000	0.66	0.0009	0.43	0.0007	0.23	1
hsa-miR-148a-5p	0.0000	0.89	0.0067	0.59	0.0054	0.30	1
hsa-miR-7155-5p	0.0002	0.96	0.0071	0.31	0.0358	0.65	1
hsa-miR-203a-3p	0.0006	1.54	0.0716	2.07	0.0066	−0.54	1
hsa-miR-6857-3p	0.0071	2.67	0.0316	0.49	0.6878	2.18	1
hsa-miR-4274	0.0055	3.09	0.0279	1.23	0.6318	1.86	1

PPGL: pheochromocytoma-paraganglioma; PPI: proton pump inhibitor treatment; pNET: pancreatic neuroendocrine tumor, log2FC: log2fold change (counts); post-hoc *p*-value: *p*-value following Tukey post-hoc test adjusted by Benjamini–Hochberg method for multiple testing correction.

**Table 2 cancers-12-02488-t002:** CgA correlation with miRNA expression in the validation set (CgA vs. –dct).

**All Samples**			
Correlation	Sample Number	Spearman	*p*-value
CgA & hsa-let-7b-5p	74	−0.29	0.0125
CgA & hsa-let-7i-5p	74	−0.46	0.0000
CgA & hsa-miR-143-3p	74	−0.34	0.0029
CgA & hsa-miR-30d-5p	74	−0.39	0.0007
CgA & hsa-miR-451a	74	−0.50	0.0000
CgA & hsa-miR-486-5p	74	−0.39	0.0006
**Healthy Controls (with & without PPI)**			
Correlation	Sample number	Spearman	*p*-value
CgA & hsa-let-7b-5p	29	−0.27	0.1599
CgA & hsa-let-7i-5p	29	−0.42	0.0249
CgA & hsa-miR-143-3p	29	−0.40	0.0293
CgA & hsa-miR-30d-5p	29	−0.39	0.0377
CgA & hsa-miR-451a	29	−0.40	0.0334
CgA & hsa-miR-486-5p	29	−0.17	0.3852
**PPGL**			
Correlation	Sample number	Spearman	*p*-value
CgA & hsa-let-7b-5p	20	−0.05	0.8256
CgA & hsa-let-7i-5p	20	−0.42	0.0655
CgA & hsa-miR-143-3p	20	−0.37	0.1069
CgA & hsa-miR-30d-5p	20	−0.35	0.1317
CgA & hsa-miR-451a	20	-0.38	0.1009
CgA & hsa-miR-486-5p	20	-0.45	0.0451
**pNET**			
Correlation	Sample number	Spearman	*p*-value
CgA & hsa-let-7b-5p	25	−0.04	0.8380
CgA & hsa-let-7i-5p	25	−0.22	0.2838
CgA & hsa-miR-143-3p	25	−0.14	0.5116
CgA & hsa-miR-30d-5p	25	−0.02	0.9157
CgA & hsa-miR-451a	25	−0.32	0.1198
CgA & hsa-miR-486-5p	25	−0.18	0.3892

**Table 3 cancers-12-02488-t003:** Diagnostic performance of CgA, individual miRNAs, and the best combination of CgA with miRNAs between different groups. *: Cut off values of CgA in ng/mL and of miRNAs in dCt; **: Cut-off in binary logistic regression model. AUC: area under curve.

PPGL vs. Controls with & without PPI treatment		AUC	Cutoff *	Sensitivity %	Specificity %	*p*-value
CgA		0.890	>102.3	95.0	72.4	<0.0001
Individual miRNAs	hsa-let-7b-5p	0.707	>12.5	90.0	48.3	0.0147
	hsa-let-7i-5p	0.710	>5.7	100.0	3.4	0.0135
	hsa-miR-143-3p	0.598	>7.7	100.0	27.6	0.2463
	hsa-miR-30d-5p	0.643	>9.9	65.0	65.5	0.0914
	miR-451a	0.678	>5.8	45.0	86.2	0.0353
	miR-486-5p	0.660	>7.3	30.0	96.6	0.0586
**Binary logistic regression model**		**AUC**	**cut-off ****	**Sensitivity %**	**Specificity %**	***p*** **-value**
Best of all combinations	CgA+hsa-let-7b-5p+hsa-miR-143-3p+hsa-miR-486-5p	0.862	0.4	85.0	82.8	0.0085
**pNET vs. Controls with & without PPI treatment**		**AUC**	**cut-off ***	**Sensitivity %**	**Specificity %**	***p*** **-value**
CgA		0.672	>102.5	64.0	72.4	0.0308
Individual miRNAs	hsa-let-7b-5p	0.702	>12.9	80.0	62.1	0.0111
	hsa-let-7i-5p	0.661	>7.9	80.0	51.7	0.0433
	hsa-miR-143-3p	0.661	>9.0	100.0	41.4	0.0433
	hsa-miR-30d-5p	0.675	>9.7	76.0	62.1	0.0276
	miR-451a	0.692	>2.9	96.0	41.4	0.0159
	miR-486-5p	0.712	>4.2	88.0	58.6	0.0078
**Binary logistic regression model**		**AUC**	**cut-off****	**Sensitivity %**	**Specificity %**	***p*** **-value**
Best of all combinations	CgA+hsa-let-7b-5p+hsa-let-7i-5p+hsa-miR-143-3p+hsa-miR-30d-5p	0.752	0.5	75.9	68.0	0.0351
**CgA low pNET vs. Controls with & without PPI treatment**		**AUCa**	**cut-off ***	**Sensitivity %**	**Specificity %**	***p*** **-value**
CgA		0.613	<79.6	100.0	37.9	0.3112
Individual miRNAs	hsa-let-7b-5p	0.651	>12.9	77.8	62.1	0.1751
	hsa-let-7i-5p	0.517	<8.8	100.0	20.7	0.8772
	hsa-miR-143-3p	0.646	>9.9	100.0	48.3	0.1920
	hsa-miR-30d-5p	0.577	>9.4	77.8	55.2	0.4923
	miR-451a	0.527	>2.9	88.9	41.4	0.8101
	miR-486-5p	0.561	>4.2	66.7	58.6	0.5828
**Binary logistic regression model**		**AUC**	**cut-off ****	**Sensitivity %**	**Specificity %**	***p*** **-value**
Best of all combinations	CgA+hsa-let-7b-5p+hsa-let-7i-5p+hsa-miR-143-3p+hsa-miR-30d-5p+hsa-miR-486-5p	0.904	0.55	66.7	96.6	0.0342

**Table 4 cancers-12-02488-t004:** Comparison of differentially expressed miRNAs in pNET/PCC tumor tissues vs. serum. miRNAs common in both tissues and sera samples are indicated in the table.

pNET					
miRNA	Corrected *p*-Value	Controls	pNET	Reference	
		avg. UMI Read Number			
miR-103a-3p	0.0371	109	65	Roldo et al., 2006 [33]	overexpressed in pNET+PACC vs. NP
				Zimmermann et al., 2018 [35]	negative correlation with Ki-67 in GEP-NET
miR-26b-5p	0.0423	115	73	Roldo et al., 2006 [33]	overexpressed in pNET+PACC vs. NP
miR-143-3p	0.0240	110	43	Jiang et al., 2015 [29]	overexpressed in INS vs pancreatic islet
miR-451a	0.0257	198	74	Jiang et al., 2015 [29]	overexpressed in INS vs pancreatic islet
miR-25	0.0153	383	138	Zimmermann et al., 2018 [35]	overexpressed in nodal met. vs. primary GEP-NET
miR-425-5p	0.01781	111	47	Zimmermann et al., 2018 [35]	overexpressed in nodal met. vs. primary GEP-NET
miR-93-5p	0.0240	286	117	Grolmusz et al., 2018 [34]	higher expression in higher grade pNET
miR-16-5p	0.0320	9670	3475	Zimmermann et al., 2018 [35]	negative correlation with Ki-67 in GEP-NET
miR-203a-3p	0.0715	19	56	Roldo et al., 2006 [33]	overexpressed in INS vs NF-pNET
**PCC**					
**miRNA**	**Corrected *p*-Value**	**Controls**	**PCC**	**Reference**	
		**avg. UMI Read Number**			
miR-101-3p	0.0273	209	56	Zong et al.2015 [36]	higher expression in malignant vs. benign PCC higher expression in *SDHD* mutation-associated tumors
				Patterson et al., 2012 [27]	differentially expressed between malignant vs. benign PCC
miR-16-5p	0.0139	9670	2423	Meyer-Rochow et al., 2010 [37]	underexpressed in malignant vs. benign PCC
miR-451a	0.005	198	39	Meyer-Rochow et al., 2010 [37]	underexpressed in malignant vs. benign PCC

**Table 5 cancers-12-02488-t005:** Patient characteristics. The following paired samples are from the same healthy control with and without PPI treatment: 45 + 46; 47 + 48; 67 + 68; 33 + 49; 69 + 70.

Sample ID.	Sex	Age at Sampling	Group	CgA (ng/mL; Ref: 19.4–98.1)	PPGL: Benign/Malignant; pNET: Grade G1-G2-G3	Experiment
51	F	12	PPGL (PCC)	548.6	benign (pr)	NGS & qPCR
52	M	41	PPGL (PCC)	10870	benign (pr)	NGS & qPCR
53	M	51	PPGL (*SDHB* mutation associated. PCC; no other malignant tumor yet)	863.4	benign (pr)	NGS & qPCR
18	F	47	PPGL (*MEN2* assoc. PCC)	402.9	malignant (rec & met)	qPCR
19	F	45	PPGL (PCC)	128.1	malignant (rec)	qPCR
23	F	67	PPGL (PCC)	749.6	benign (pr)	qPCR
20	M	56	PPGL (PGL)	8166	malignant (rec & met)	qPCR
56	M	40	PPGL (*SDHB* mutation associated PGL)	2265	malignant (rec & met)	NGS & qPCR
57	F	69	PPGL (PCC)	1375	malignant (rec & met)	NGS & qPCR
21	F	44	PPGL (PGL)	273	malignant (rec & met)	qPCR
62	F	48	PPGL (PCC)	2409	benign (pr)	NGS & qPCR
28	M	20	PPGL (PCC)	809.5	NA	qPCR
22	F	35	PPGL (*NF1* mutation associated PCC; no other malignant manifestation)	1602	malignant (rec & met)	qPCR
64	F	35	PPGL (PCC)	487.9	benign (pr)	NGS & qPCR
26	M	24	PPGL (*VHL* mutation associated PCC; no other malignant tumor)	63.2	malignant (bilateral)	qPCR
24	F	69	PPGL (PCC)	2295	malignant (rec & met)	qPCR
29	M	21	PPGL (PCC)	104.1	benign (pr)	qPCR
27	F	62	PPGL (PCC)	115	benign (pr)	qPCR
66	F	15	PPGL (SDHB mutation associated PCC)	470.8	malignant (rec & met)	NGS & qPCR
25	M	36	PPGL (PCC)	1037.4	malignant (rec & met)	qPCR
1	M	75	pNET	34.2	G1	qPCR
54	F	58	pNET	191.5	NA	NGS & qPCR
55	M	67	pNET	176.6	G3	NGS & qPCR
2	M	57	pNET	35.2	G2	qPCR
3	F	62	pNET	62.2	G2	qPCR
4	M	77	pNET	67.9	G2	qPCR
58	F	66	pNET	162.7	NA	NGS & qPCR
59	M	52	pNET	448.5	G2	NGS & qPCR
5	F	62	pNET	44.2	G2	qPCR
60	M	72	pNET	1679	G2	NGS & qPCR
61	F	61	pNET	543.6	G2	NGS & qPCR
6	M	39	pNET	327.4	G3	qPCR
63	F	47	pNET	1009	G2	NGS & qPCR
7	M	68	pNET	219.2	G2	qPCR
65	M	74	pNET	104.5	G1	NGS & qPCR
8	M	42	pNET	190.1	G3	qPCR
9	F	48	pNET	115.4	NA	qPCR
10	F	82	pNET	79.1	G1	qPCR
11	F	76	pNET	132	G2	qPCR
12	F	58	pNET	51.5	G1	qPCR
13	M	75	pNET	2490.8	NA	qPCR
14	F	64	pNET	4163	G2	qPCR
15	F	59	pNET	275.7	NA	qPCR
16	M	72	pNET	49.8	G2	qPCR
17	M	73	pNET	34.4	NA	qPCR
74	M	77	healthy control	38.5	na	qPCR
38	M	71	healthy control	53.7	na	qPCR
40	F	49	healthy control	12.7	na	qPCR
31	M	49	healthy control	41.5	na	qPCR
76	M	44	healthy control	39.8	na	NGS & qPCR
36	F	68	healthy control	100.4	na	qPCR
30	F	13	healthy control	18	na	qPCR
34	M	40	healthy control	40.2	na	qPCR
35	F	69	healthy control	48.7	na	qPCR
41	F	47	healthy control	93	na	qPCR
39	F	63	healthy control	55	na	qPCR
75	M	65	healthy control	39	na	qPCR
37	M	55	healthy control	57.2	na	qPCR
73	M	47	healthy control	34	na	NGS & qPCR
43	F	16	healthy control	52.6	na	qPCR
42	F	33	healthy control	68.4	na	qPCR
72	F	46	healthy control	42.8	na	NGS & qPCR
32	F	58	healthy control	80.2	na	qPCR
71	M	37	healthy control	27.7	na	NGS & qPCR
45	F	72	healthy control without PPI	494	na	qPCR
46	F	72	healthy control on PPI	5775	na	qPCR
47	F	62	healthy control without PPI	198.9	na	qPCR
48	F	62	healthy control on PPI	516	na	qPCR
67	F	39	healthy control without PPI	42.3	na	NGS & qPCR
68	F	39	healthy control on PPI	197.8	na	NGS & qPCR
33	M	64	healthy control without PPI	186.7	na	qPCR
49	M	64	healthy control on PPI	595.4	na	qPCR
69	F	46	healthy control without PPI	45.6	na	NGS & qPCR
70	F	46	healthy control on PPI	278	na	NGS & qPCR

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
