# Peer review of "Circulating miRNA Increases the Diagnostic Accuracy of Chromogranin A in Metastatic Pancreatic Neuroendocrine Tumors"

_cancers, 2020, doi:10.3390/cancers12092488_

Round 1

Reviewer 1 Report

In this article the authors investigate the diagnostic value of circulating miRNAs along with Chromogranin A (CgA) in pancreatic neuroendocrine tumors (pNET). Next-generation sequencing was used to determine miRNA expression profiles in 74 serum samples from patients with pNET, pheochromocytoma/paraganglioma, healthy individuals with normal CgA, and healthy individuals with and without current PPI treatment.

Patients with NET showed a global downregulation of miRNAs compared to healthy people in this study. Out of 1525 miRNAs detected in NGS profiling, 33 miRNAs were able to discriminate tumor samples from controls. None of these miRNAs ALONE was found to be applicable as a biomarker for pNET or pheochromocytoma/paraganglioma. Interestingly, the authors were able to show that the predictive value of Chromogranin A in combination with a set of miRNAs could be significantly improved.

They conclude that a set of miRNAs increased the diagnostic utility of CgA in pNET even in patients with low CgA.

Overall, this study is professionally written in terms of language and content. The methods used are appropriate and the conclusions drawn are reasonable. The authors have put a lot of effort into processing detailed questions, without losing the clear overview. The topic is of high clinical relevance.

Accept as it is for publication.  

Author Response

Thank you very much for reviewing our manuscript. We were especially delighted to learn that you found our paper professionally written and of high clinical relevance.

Reviewer 2 Report

The main limitation of the study is the sample size. Too small to arrive at effective and robust conclusions to provide precise indications in the use of markers in NENs. But the data in scientific literature relating to the determination of markers in NENs with miRNA and NSG are very few.

The introduction must be expanded on a more extensive discussion of the markers related to the NENs. It would be useful to also describe the data relating to NSE and 5HIAA

it would be useful to better clarify  the concepts from line 227 to line 230

Author Response

We are grateful the Reviewer for evaluating our manuscript, and we also appreciate your relevant and precise comments made on it. Below, please find our detailed response to your comments.

  1. The introduction must be expanded on a more extensive discussion of the markers related to the NENs. It would be useful to also describe the data relating to NSE and 5HIAA

Answer: We thank the Reviewer for his/her suggestion. Upon the valuable request, we expanded the introduction with highlighting the importance of additional biomarkers as NSE and HIAA besides miRNAs. Please find highlighted in the Introduction section.

  1. It would be useful to better clarify the concepts from line 227 to line 230

Answer: We grateful the Reviewer for his/her suggestion for clarification. As different miRNA signatures have been described in malignant vs. benign (Patterson et al. 2012) and sporadic vs. hereditary PPGL (Tömböl et al. 2010) we aimed to investigate serum miRNAs in that regard as well. Now, we clarified this in the Result section, please find it highlighted (from line 252).

Reviewer 3 Report

This manuscript analyzed the serum-miRNA profile in pNETs and PPGLs with the aim of discriminating pNET/PPGL samples from healthy controls. I find several issues that should be clarified.

  1. The authors perform an “exploratory” study by NGS and subsequently, a validation study in an an extended sample cohort. The manuscript includes data of the validation cohort but do not indicate the patient characteristics of their first exploratory analysis. If the validation was experimental, in the same samples as the exploratory study, the authors should clarify this and change the manuscript accordingly.
  2. The rationale for considering the PPI treatment should be better explained in the introductory section of the manuscript.
  3. The authors conclude, from their analysis that: In the pNET group compared to CgA the use of individual let-7b-5p, miR-30d-5p, miR-451a and miR-486-5p in ROC analysis resulted in better discrimination (Table 3). However, the data in CgA vs combination (AUC :0.67 vs 0,75; sensitivity: 64% vs 75.9%; specificity: 72.4% vs 68%) are only slightly better in AUC and sensitivity, specificity decreases. The authors should discuss about the robustness of their data and conclusions. Based on the data the authors have thus far, do they think it would be worthy to perform further study in a larger cohort of patients to translate their findings into clinically useful data?
  4. In the results section, the authors do not mention Figure 3B.
  5. Section pNET/PPGL tumor tissue miRNAs in serum has a confusing title. In addition, I honestly think that the section does not add further value to the study and could be omitted.
  6. Figure 2 is not sufficiently explained: it should mention the data on healthy controls with PPI. Again, the rationale of considering PPI treatments in this study should be explained.
  7. It is confusing the terms healthy individuals, healthy controls… For example, I recommend rephrasing sentences as for example: Line 228: we found that in pNET samples CgA alone was less effective in discrimination of pNET sera from healthy individuals compared to healthy controls. In Table 2, are authors include Healthy control (19 samples), Healthy control without PPI (5 samples) and Healthy control on PPI (5 samples). Please explain better the mesning of those controls, especially the difference between Healthy control and Healthy control without PPI.
  8. References regarding CgA sensitivity in gastrinomas, pheochromocytomas and nonfunctioning pNETs (1,6,7; line 44) are revisions; studies specifically analyzing the value of CgA in these patients should be referenced.
  9. In the discussion section, I think that the authors should focused on the value of CgA versus miRNA; first two paragraphs of the discussion are less relevant in this context. The data on PPGLs discriminating whether patients have or not germline mutations, given the reduced number of cases, does not allow to reach useful conclusions.

Author Response

We are grateful to the Reviewer for his/her precise comments and valuable criticism. Below, please find detailed response to your observations and requests.

  1. The authors perform an “exploratory” study by NGS and subsequently, a validation study in an extended sample cohort. The manuscript includes data of the validation cohort but do not indicate the patient characteristics of their first exploratory analysis. If the validation was experimental, in the same samples as the exploratory study, the authors should clarify this and change the manuscript accordingly.

Answer: We are especially grateful the Reviewer for his/her observation to describe the discovery and validation cohorts in a clearer way. Upon your valuable requests, we clarified this issue: first, in the Patients and controls section of Materials and methods, we detailed the data on controls; second, in Table 1 we indicated exactly which samples were included in the discovery and which ones in the validation cohort; and third, in the Results part (in the “Analysis of circulating miRNAs by next generation sequencing” and in the “miRNA validation” section) we specified the exact number of samples. Please find these highlighted in the revised version of the manuscript.

  1. The rationale for considering the PPI treatment should be better explained in the introductory section of the manuscript.

Answer: We thank the Reviewer for his/her suggestion. Several studies so far, including one from our research group, highlighted, that PPI administration results in a significant increase of CgA, thus decreasing the discriminatory power of CgA as a biomarker for NETs. Therefore, we detailed the background for considering PPI treatment in relation to the diagnostic value of CgA. Please find the clarification and the references in the Introduction paragraph highlighted (from line 61).

  1. The authors conclude, from their analysis that: In the pNET group compared to CgA the use of individual let-7b-5p, miR-30d-5p, miR-451a and miR-486-5p in ROC analysis resulted in better discrimination (Table 3). However, the data in CgA vs combination (AUC :0.67 vs 0,75; sensitivity: 64% vs 75.9%; specificity: 72.4% vs 68%) are only slightly better in AUC and sensitivity, specificity decreases. The authors should discuss about the robustness of their data and conclusions. Based on the data the authors have thus far, do they think it would be worthy to perform further study in a larger cohort of patients to translate their findings into clinically useful data?

Answer: We appreciate the Reviewer profoundness regarding this question. We completely agree with the Reviewer’s opinion that the small sample size may be a limitation of our study. However, using our data from NGS analysis, the number of patients and controls of our validation cohort was calculated with sample size calculator, using a statistical power >80% and p<0.05 (http://powerandsamplesize.com/Calculators/). In our validation phase, we used more samples in each comparison than the calculation showed. Also, pNET (especially, only a subset of it: nonfunctioning NET) is a rare tumour type, and in every day clinical practice often is diagnosed only when already metastasized. Therefore, obtaining samples before surgery and diagnosis are challenging, In line with this, there are only a few studies (evaluating limited samples too) showing data about circulating miRNA profile of pNET patients.

We truly believe that further studies in larger cohorts could be useful, especially, including samples before and after treatment/surgery. 

  1. In the results section, the authors do not mention Figure 3B.

Answer: Description of Figure 3B can be found in the original manuscript, in the second paragraph of section „Association of miRNA expression with clinicopathological parameters” of the Results, i.e. „Investigating miRNAs in relation to the tumor grade, in pNET patients a tendency of inverse correlation between miRNS: let-7i-5p, miR-30d-5p and miR-451a and grade was observed (Figure 3B)” (line 226). However, this was modified in the revised version to line 250. Please find it highlighted.

  1. Section pNET/PPGL tumor tissue miRNAs in serum has a confusing title. In addition, I honestly think that the section does not add further value to the study and could be omitted.

Answer: We thank the Reviewer for drawing our attention to the confusing title. In other tumours (such as adrenocortical cancer) deregulated miRNAs can be detectable in circulation. Hence, these miRNAs can be considered tissue-specific miRNAs originating directly from the tumour tissue itself. In this paragraph we intended to analyze tumour tissue-derived miRNAs in the circulation. Therefore, we collected tissue-based miRNA studies and we cross-referenced these findings with our results. By doing this analysis we identified miR-203a-3p as overexpressed both in pNET tissue and serum indicating that this miRNA can be originated directly from pNET tissue. In PPGL, miR-16-5p and miR-451a as putative tumour suppressor miRNAs were downregulated in both tissue and serum.

Upon your comment, we changed the title of this section for better understanding, also as further clarification, please find the changes highlighted.

  1. Figure 2 is not sufficiently explained: it should mention the data on healthy controls with PPI. Again, the rationale of considering PPI treatments in this study should be explained.

Answer: We are grateful this observation of the Reviewer. We detailed data on controls in the Results section regarding Figure 2, as well as in Figure Legends. We refer to comment 2. regarding the explanation of the rationale of considering PPI treatment. Please find all additional explanations highlighted.

  1. It is confusing the terms healthy individuals, healthy controls… For example, I recommend rephrasing sentences as for example: Line 228: we found that in pNET samples CgA alone was less effective in discrimination of pNET sera from healthy individuals compared to healthy controls. In Table 2, are authors include Healthy control (19 samples), Healthy control without PPI (5 samples) and Healthy control on PPI (5 samples). Please explain better the mesning of those controls, especially the difference between Healthy control and Healthy control without PPI.

Answer: We thank the Reviewer for his/her suggestion to improve the quality of our manuscript. We clarified the composition of the control group in the Methods and Results section (referring to the Reviewer’s previous request). Among all controls, we included 5 healthy volunteers that were treated with PPI as described in the Materials and methods. Samples collected before and after PPI treatment were included in the study. To differentiate these paired samples, we refer to them as „healthy control without PPI” and „healthy control on PPI”. The rest of the controls did not take PPIs („healthy controls”).

In Table 2 the “control&PPI” group includes all healthy control with and without PPI treatment. The goal of using this merged group was to discriminate pNET and PPGL patients from “tumour-free” healthy individuals irrespective of the usage of PPI. Upon your valuable request, we clarified these details in the table captions of Table 1 and Table 2.

  1. References regarding CgA sensitivity in gastrinomas, pheochromocytomas and nonfunctioning pNETs (1,6,7; line 44) are revisions; studies specifically analyzing the value of CgA in these patients should be referenced.

Answer: We appreciate the Reviewer’s valuable comment. According to this, we completed the reference list with an adequate paper presenting data about the value of CgA (Nobels FR, Kwekkeboom DJ, Coopmans W, et al. Chromogranin A as serum marker for neuroendocrine neoplasia: comparison with neuron-specific enolase and the alpha-subunit of glycoprotein hormones. J Clin Endocrinol Metab. 1997;82(8):2622-2628. doi:10.1210/jcem.82.8.4145).

  1. In the discussion section, I think that the authors should focused on the value of CgA versus miRNA; first two paragraphs of the discussion are less relevant in this context. The data on PPGLs discriminating whether patients have or not germline mutations, given the reduced number of cases, does not allow to reach useful conclusions

Answer: We are grateful for the Reviewer’s suggestion. In the first two paragraphs of the Discussion we aimed to point out the importance of research on circulating biomarkers in neuroendocrine tumours. Very limited diagnostic tools are available for this tumour entity, therefore our study may facilitate others to clarify whether circulating miRNAs can be useful for these tumours too To date, no data on global miRNA abundance in blood of patients with neuroendocrine tumours have been presented. Based on our findings, the overall miRNA level in blood can be a general indicator of the presence of a neuroendocrine tumour, similarly to other solid tumours presented earlier (reference 23).

Also, we totally agree, that the small sample size does not allow drawing solid conclusion. However, due to the rarity of this tumour type, there is no information on the association on PPGL mutation status and circulating miRNAs, therefore we think that this is still an important finding to report, giving opportunity to other groups to investigate this on their own sample cohort.

Round 2

Reviewer 3 Report

The authors provided correct answer to all my remarks. It is acknowledged the work performed in a rare tumor and this outperforms the limitation of the study in a rather small sample size.

There is still an issue that I would like to be clarified: the authors indicate that the accuracy of diagnosis of CgA and a set of miRNA in serum of patients is high even though none of these parameters are modified in comparison with normal serums?

Please clarify sentence in line 188: Parentheses after accuracy% represents 95% confidence intervals. I have not found these data in the Table.

There are some spelling errors and sentences that need to be rephrased (line 147, lines 191-193).

Author Response

Comments and Suggestions for Authors – Reviewer #3

 The authors provided correct answer to all my remarks. It is acknowledged the work performed in a rare tumor and this outperforms the limitation of the study in a rather small sample size.

  1. There is still an issue that I would like to be clarified: the authors indicate that the accuracy of diagnosis of CgA and a set of miRNA in serum of patients is high even though none of these parameters are modified in comparison with normal serums?

Answer: The Reviewer raised an essential point. In comparison of pNET or PPGL groups with controls, levels of CgA and miRNAs showed significant differences (please find in Table 4). If samples from controls treated with PPI were included either CgA or miRNA signature alone was not applicable as biomarker for PPGL/pNET. In addition, neither CgA nor any miRNA alone was able to discriminate low CgA pNET patients from controls treated with PPI. However, the combination of CgA concentration and expression level of miRNAs in a regression model yielded higher AUC to those obtained using individual markers, demonstrating that our model was effective in distinguishing different groups in every comparison. Our results are in line with previous findings showing that in other cancer types, independently of individually being significantly different or not, combining more factors in a regression model increases the discriminatory value of biomarkers. (Liu et al. 2011 Combination of plasma microRNAs with serum CA19‐9 for early detection of pancreatic cancer; or Pesta et al. 2019. Plasma microRNA Levels Combined with CEA and CA19-9 in the Follow-Up of Colorectal Cancer Patients.)

This clarification has been included into the Conclusion section. Please find it highlighted.

We thank again the Reviewer for asking for clarification

  1. Please clarify sentence in line 188: Parentheses after accuracy% represents 95% confidence intervals. I have not found these data in the Table.

Answer: We completely agree with the Reviewer. This sentence remained from a previous version of the manuscript. We apologize for the mistake and we deleted the misleading sentence.

  1. There are some spelling errors and sentences that need to be rephrased (line 147, lines 191-193).

Answer: We are grateful the Reviewer for his/her correcting suggestions.

Line 147: misspelling was corrected.

Line 191-193: Sentences were rephrased as follows:

Please find highlighted the changes in the revised manuscript.